# Crude Oil Price Movements and Institutional Traders

**Celso Brunetti** [1] , **Jeffrey H. Harris** [2,*] **and Bahattin Büyükşahin** [3]

1   Federal Reserve Board, 20th and C Streets NW, Washington, DC 20551, USA; celso.brunetti@frb.gov
2   Kogod School of Business, American University, 4400 Massachusetts Ave NW, Washington, DC 20016, USA
3   CoMeX Consulting and Advising, 4105 N Ridgeview Rd, Arlington, VA 22207, USA;
    bahattin.buyuksahin@gmail.com
*   Correspondence: jharris@american.edu; Tel.: +1-202-885-6669

**Abstract:** We analyze the role of hedge fund, swap dealer, and arbitrageur activity in the crude oil market. The contribution of our work is to examine the role of institutional traders in switching between high-volatility and low-volatility regimes. Using confidential position data on institutional investors, we first analyze the linkages between trader positions and fundamentals. We find that these institutional position changes reflect fundamental economic factors. Subsequently, we adopt a Markov regime-switching model with time-varying probabilities and find that institutional position changes contribute incrementally to the probability of regime changes.

**Keywords:** trader positions; fundamentals; price reversals

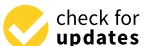



## 1. Introduction

Recent episodes of commodity price changes have rekindled the debate about whether speculative activity affects commodity prices. A significant body of work shows that fundamental supply and demand affect market prices (see, for instance, [1–16]), while other studies argue that excessive speculation can lead prices to depart from fundamentals (see, for instance, [17–24]). Some argue that "massive passive" index investors or rampant speculation by hedge funds (both examples of "financialization" in commodity markets) have created excessive volatility or irrational prices. Empirically, it is difficult to disentangle fundamental effects from financialization effects, since all traders, including speculators, likely condition trades on fundamental supply and demand. In this paper, we exploit detailed daily trading data to isolate non-fundamental speculative activity and relate this activity to the continuation or reversal of price trends.

In this study, we find broad evidence that macroeconomic announcements and news affect all types of trader position changes—including commercial traders (merchants, manufacturers, and producers) as well as speculative traders (swap dealers and hedge funds). Trading activity not explained by fundamentals likely reflects market expectations, private beliefs (speculative or otherwise), and/or private information of these traders. We explore whether and how the unexplained component of trading relates to crude oil price dynamics net of fundamental effects.

Empirically, we identify detailed trader positions from the CFTC's Large Trader Reporting System (the CFTC audits the data and produces weekly public Commitments of Traders for four trader groups (*producer/merchants*, *swap dealers*, *managed money traders/hedge funds*, and *other non-commercials*). Our data identify producer, manufacturer, dealer, swap dealer, hedge fund, floor trader, arbitrageur, and non-reportable subcategories) and apply Markov switching models (conditioning on unexplained—by fundamentals—trader positions) as a systematic approach to modeling futures price data. We recursively generate daily probabilities in the model to allow for regime shifts in the data-generating process

(Markov regime-switching models can also capture fat tails, asymmetries, autocorrelation, volatility clustering, and mean reversion in financial asset series (see [25])). Many authors argue that nonlinear processes model the behavior of financial variables better than linear processes—e.g., [26–28]. We find the Markov switching approach accommodates the linkages between unexplained institutional position changes and price trends and reversals.

The existence of different market regimes has important implications for market regulators, portfolio managers, and liquidity providers alike. Market regulators concerned about long-term trends, reversals, and bubbles in market prices might more effectively implement policy choices with a better understanding of regimes and the determinants of regime switching. Portfolio managers can adopt regime-dependent strategies to maximize risk-adjusted returns as well ([29] note that regime-switching strategies can be defined by distributions of regime-dependent returns, exposure to underlying risk factors, and/or alphas. Hedge funds are perhaps most likely to implement dynamic switching strategies such as a long/short equity strategy). In addition, liquidity providers who learn from order flow can more effectively manage inventories with better information about the transition probability of regime changes.

The remainder of the paper is organized as follows. Section 2 discusses the related literature and in Section 3 we describe the data in detail. Section 4 analyzes the relationship between institutional investor positions and fundamentals. Section 5 presents the Markov regime-modeling strategy we adopt and discusses the main results. Section 6 concludes the paper.

## 2. Literature Review and Discussion

Our approach is in line with other studies. Ref. [30] assume that price reversals cannot be generated by a single trader, but rather by coordination among rational traders. Importantly, non-commercial institutions potentially bring market-moving information to oil markets. Hedge funds, for instance, apply complicated modeling techniques and use proprietary valuation models to take both short and long futures positions. Likewise, swap dealers bring distilled order flow from knowledgeable clients (via over-the-counter (OTC) swaps and index trader demand) into WTI futures as well.

Notably, both hedge funds and swap dealers have gained market share concurrent with recent periods of large oil price swings. Hedge fund market share quadrupled from about 7 to 28 percent while swap dealer market share rose from about 35 to 40 percent from 2000 to 2008 [31] (commodity index fund positions continued to grow from USD 9 billion to almost USD 300 billion from 2000 to 2012 according to the CFTC's Index Investment Data).

The crude oil futures market reflects all three aspects of [30,32] synchronization risk (Crude oil, for instance, rose from USD 32 per barrel in 2003 to over USD 145 in July 2008 before falling to USD 35 by December 2008 during our sample period (see Figure 1)). First, the WTI futures market (with over 1 million open interest contracts since mid-2004) is suitably large enough that a single arbitrageur is unlikely to correct mispricing. Second, the WTI market enjoys a significant number of competitive, rational arbitrageurs (e.g., hedge funds) that, given the complexities of determining worldwide supply and demand, are likely to become sequentially aware of any price deviation from fundamental value. Third, both long and short positions in futures markets expose hedge funds to significant holding costs, given the real cash flows associated with mark-to-market margins.

By aggregating traders by type—hedge funds, commodity index traders, etc.—our tests shed light on which group prompts regime switches in price trends and volatility. Using the unexplained component of trading, we find that hedge fund position changes are significantly related to crude oil price trend reversals.

Ref. [30] also introduce a temporal dimension to the coordination problem which assumes that arbitrageurs receive information sequentially. We test this temporal dimension by examining the lag (in days) between aggregate position changes for different trader types and subsequent price reversals. We find that crude oil prices and volatility react to hedge fund trading with a delay of a single day.

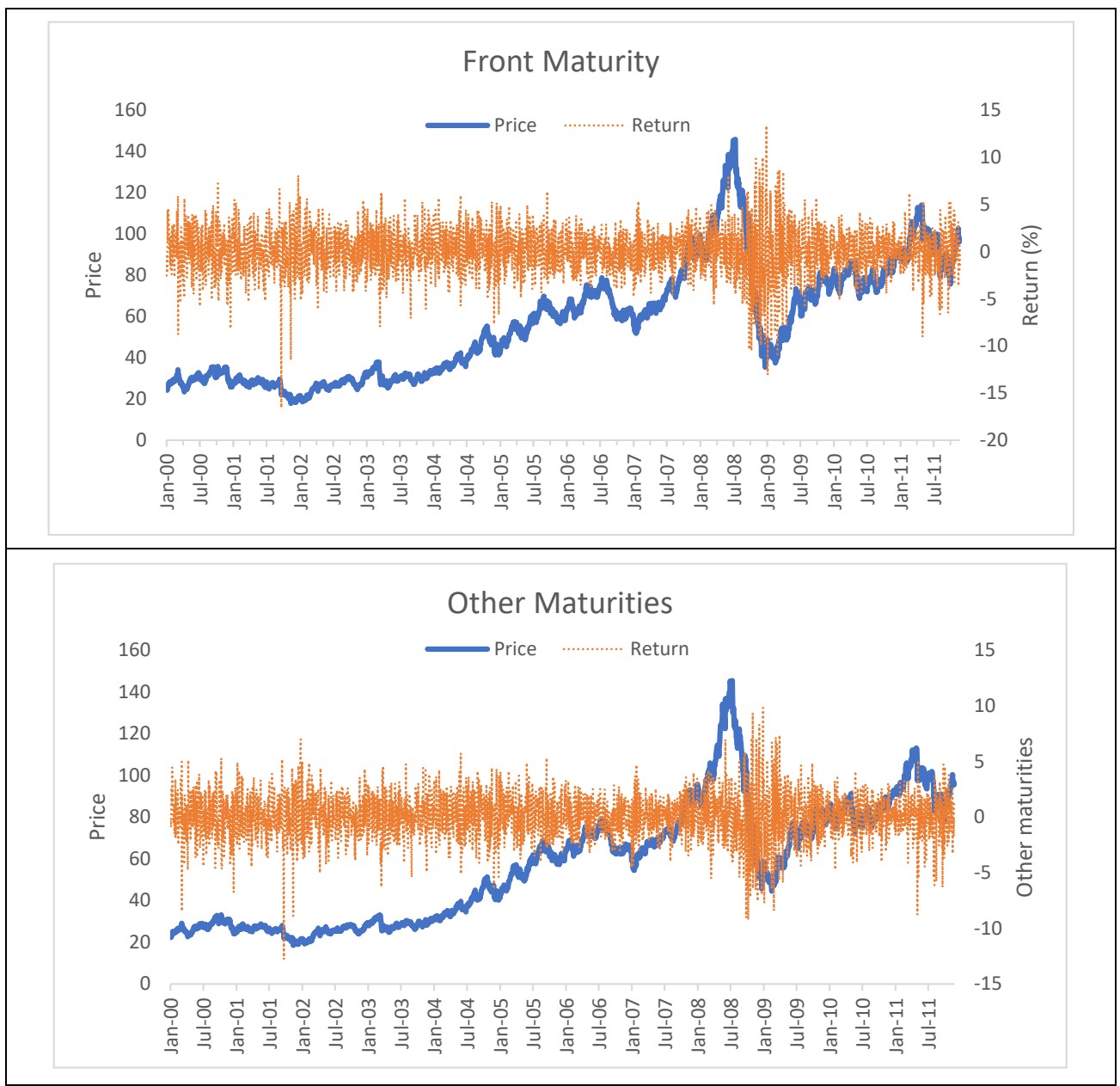

**Figure 1.** Prices and returns.

Notably, unexplained (by fundamentals) aggregate position changes of merchants, manufacturers, producers, and floor brokers contribute significantly to price continuations but not to reversals. Swap dealer activity is largely unrelated to either continuations or reversals, likely reflecting their passive aim of gaining long oil market exposure. Therefore, the "massive passive" index fund flows executed via swap dealers have no discernable effect on crude oil price continuations or reversals.

Our work builds on [33], who identify macroeconomic announcements that affect WTI crude oil prices. We consider these announcements together with several other macroeconomic variables. Similar to their results, we confirm strong linkages between institutional trader positions and crude oil market fundamentals.

Our work is also related to the growing literature on crude oil prices and volatility. Ref. [12] develop a structural model where they show that speculative shocks have no impact on oil prices. Likewise, [15,16,34] find little evidence that "financialization" or speculation over the past decade led to excessive volatility or prices that deviated from

fundamentals ([35] finds the oil price/inventory relation is stable over time indicating that the effects of financialization are muted. Ref. [36] also analyze oil inventories. Ref. [37] review the literature on speculation in oil markets. See also [38,39]).

Other studies purport to demonstrate that speculation or "financialization" either holds the potential for or actually leads to excessive volatility and/or prices decoupled from fundamentals (e.g., [19,21,40]; among others) ([40] estimates index fund positions imputed by [41], but ref. [8] show that this computation is wrong. Ref. [13] show that Singleton's results only hold for a very short period and are most likely driven by increased correlations during the crisis. Others document increased correlations between commodities and other markets, particularly during the crisis—e.g., [17,20]). Our findings that institutional trader groups contribute to price reversals add to our understanding of how institutions affect markets. In fact, our results suggest that synchronization risk [30,32] plays a role in oil market price reversals, consistent with institutional trading driving the rise and fall of tech stocks at the turn of the century [42].

## 3. Data

We analyze crude oil futures contracts, one of the most important commodity markets that has experienced significant price swings during our sample period. We collect daily futures prices and trader positions data from 5 January 2000 to 25 November 2011, concentrating on the six largest market participant categories: hedge funds, swap dealers, merchants (including wholesalers, exporters/importers, shippers, crude oil marketers, etc.), manufacturers (including fabricators, refiners, etc.), producers, and floor brokers. We analyze hedge fund positioning since hedge funds are considered some of the most sophisticated traders and are the largest (based on open interest) non-commercial trader group in this market ("non-commercial" refers to traders with limited or no interest in producing, consuming, storing, or transporting the underlying commodity). We analyze swap dealer positions since swap dealers handle both sophisticated OTC and commodity index trades. Merchants, manufacturers, and producers are the most important hedgers in the crude oil market.

Our models consider both the nearby contract (i.e., the contract closest to delivery) and all contract maturities since our goal is to capture dynamics across the entire term structure of oil contracts. Although most of the liquidity concentrates in the nearby contract, there is evidence that longer maturity contracts may contain important information (see [31]) and many longer maturity contracts trade actively on a daily basis.

Most oil market participants typically avoid delivery issues and roll over positions from the nearby contract to the next-to-nearby contract before maturity (i.e., the expiration date). This behavior generates seasonality in the position data. To mitigate seasonality, we adopt a roll-over strategy and switch to the new contract when the open interest of the nearby contract falls below the open interest of the next-to-nearby contract. In this regard, our roll-over strategy also avoids price and position changes generated by delivery considerations at or near contract expirations. When we consider all contract maturities, positions are constructed such that seasonality and delivery distortions are not influential.

### 3.1. Futures Market Return Data

We compute returns in two ways. First, we calculate daily returns based on daily settlement (closing) prices of the nearby contract as $r_t^{front} = p_t^{front} - p_{t-1}^{front}$, where $p_t^{front}$ is the natural logarithm of the settlement price in day $t$. When we switch the contract from the nearby to the next-to-nearby, $p_t^{front}$ and $p_{t-1}^{front}$ refer to the next-to-nearby contract. We refer to these as the *front month* returns. Second, we account for all other contract maturities traded on a given day and construct the daily price as the weighted average (by open interest) settlement price of each maturity contract. We refer to those prices as $p_t^{all}$ and to the returns as $r_t^{all}$ (the last week of trading of the nearby contract is always excluded in the computation of $r_t^{all}$).

Rows one and two in Table 1 report summary statistics for returns. The table presents the distribution of daily returns and trader positions for the nearby crude oil futures contracts. We switch from the nearby contract to the next-to-nearby contract when the open interest of the nearby contract falls below that of the next-to-nearby contract. Mean daily crude oil returns are positive with $r_t^{all}$ having higher daily returns (both mean and median) and lower standard deviation than $r_t^{front}$. This is to be expected since the averaging of $r_t^{all}$ effectively smooths the time series. The unconditional distribution is non-Gaussian with negative skew and kurtosis in excess of three.

**Table 1.** Summary Statistics.

| | Mean | Median | Max | Min | Std. Dev. |
|---|---|---|---|---|---|
| **Number of Observations: 2964** | | | | | |
| **Returns** (%) | | | | | |
| Front maturity | 0.017 | 0.085 | 13.345 | −16.544 | 2.365 |
| Other maturities | 0.047 | 0.092 | 9.835 | −12.782 | 1.979 |
| **Merchant** | | | | | |
| Front maturity | −50,227 | −49,178 | 32,641 | −146,601 | 35,250 |
| Other maturities | −44,904 | −30,958 | 90,551 | −206,337 | 79,448 |
| **Manufacturer** | | | | | |
| Front maturity | −20,863 | −19,730 | 14,033 | −59,984 | 12,348 |
| Other maturities | −25,529 | −26,911 | 28,789 | −64,976 | 17,583 |
| **Producer** | | | | | |
| Front maturity | −7621 | −7800 | 12,053 | −30,511 | 6638 |
| Other maturities | −9787 | −10,819 | 16,042 | −33,587 | 9191 |
| **Floor Broker** | | | | | |
| Front maturity | −1166 | −503 | 10,987 | −16,831 | 3331 |
| Other maturities | 898 | 542 | 18,658 | −21,134 | −11,640 |
| **Swap Dealer** | | | | | |
| Front maturity | 72,230 | 64,777 | 193,253 | −9695 | 39,238 |
| Other maturities | 4589 | 2261 | 102,950 | −107,205 | 56,141 |
| **Hedge Fund** | | | | | |
| Front maturity | 6087 | 6853 | 90,328 | −93,504 | 30,229 |
| Other maturities | 72,502 | 60,304 | 318,133 | −36,555 | 62,026 |

Figure 1 depicts prices and returns for the front maturity and for all other maturities with some periods of high volatility (during the 2008–2009 crisis) and low volatility evident. In particular, high volatility is associated with falling prices.

### 3.2. Market Participant Positions

The CFTC collects data on the positions of large traders that hold positions above CFTC-specified levels (during our sample period, the large trader reporting level is 350 contracts for crude oil). Total CFTC-reported trader positions represent approximately 70 to 90 percent of total open interest in the market, with the remaining open interest representing smaller (non-reportable) traders. The CFTC classifies each reporting trader based on self-reported business models.

In this paper, we concentrate on the six largest categories of market participants in the crude oil market, including commodity swap dealers and hedge funds. Although the Commodity Exchange Act does not formally define hedge funds, we classify Commodity Pool Operators, Commodity Trading Advisors, and Associated Persons who may control customer accounts as hedge funds. We also include other participants (identified by CFTC Market Surveillance staff) known to be managing money as hedge funds. We cross-reference our hedge fund list with press reports to directly confirm that these traders are considered to represent hedge funds.

In commodity markets, swap dealers play an important role, using derivative markets both to manage price exposure originating from their OTC swap business and to manage transactions with commodity index funds. Index funds are often utilized by large institutions to diversify with commodities, so commodity index funds typically hold significant long-only positions in near-term futures contracts. Over our sample, as commodity index funds have grown significantly, swap dealer positions have grown concurrently. Merchants, manufacturers, and producers are the largest hedgers in the crude oil market. Lastly, floor brokers facilitate transactions in the pit and are believed to convey important information to the market as well (open outcry trading of crude oil on all CME Group platforms ceased on 2 July 2015).

We analyze positions in both futures and options (delta-adjusted to futures equivalence) and compute the net total positions as the difference between long and short positions for both the nearby contract and for all other contracts. While we consider both futures and options positions because we are interested in the overall exposure of institutional investors in the crude oil market, our results are robust to considering only net futures positions.

Table 1 shows descriptive statistics for the positions of each trader group. Merchants hold net short positions, in line with their traditional hedging role, tilting toward nearby contracts—merchants are net short 50,000 nearby contracts compared to under 45,000 contracts at all other maturities. Like merchants, manufacturers and producers are traditional hedgers and hold net short positions. Unlike merchants, however, manufacturers and producers more commonly utilize longer-term contracts—their mean and median positions are larger for other maturities than for the front month.

As expected, floor brokers have relatively small net positions since floor brokers facilitate trading in the pit and usually do not carry large inventories. The standard deviation of floor broker positions is the largest (relative to the mean) among all trader groups, indicating that floor brokers may convey information to the market.

Swap dealers hold large net long positions in the front maturity contract but almost no net positions in longer maturities. We conjecture that swap dealers use the front month to manage their transactions with commodity index funds and the longer maturities to manage their price exposure originating from their OTC business. Lastly, hedge funds are on average net long, with much larger long positions in distant contracts than in the front month, implying that these sophisticated investors use the entire futures curve to gain exposure to the crude oil market.

Table 2 shows the participation rate of each trader category as a percentage of the total open interest broken down by long and short positions. Table 2 shows that hedge funds hold both long and short positions in approximately equal amounts. As expected, swap dealers mainly hold long positions, with nearly 40 percent of long positions in the front maturity. Merchants, manufacturers, and producers predominantly hold short positions that reach up to 47 percent of open interest.

Figure 2 depicts the quarterly participation rate of each market participant, computed as the sum of futures and options, long and short positions divided by two. Merchants have similar participation rates in the front maturity as in other maturities. Manufacturers and producers have only a fraction of their exposure in the front maturity, consistent with longer-term hedge positions.

Floor brokers have low participation rates in the front maturity but they are more active in longer maturities. Participation rates for swap dealers at the beginning of our sample are similar in both front month and distant contracts. However, starting in 2004 when CIT demand began to grow, swap dealer positions shifted toward the front month. From 2000 to 2004 hedge funds have larger exposures to longer-term maturities but thereafter concentrate exposures more on the front month. Overall, Figure 2 shows that institutional participation in the crude oil market is not always consistent over time and it is important to account for all positions across the term structure and not just those in the front maturity contract.

**Table 2.** Long/Short Percentage of Open Interest.

| | Median | | Max | | Min | |
|---|---|---|---|---|---|---|
| | **Long** | **Short** | **Long** | **Short** | **Long** | **Short** |
| **Merchant** | | | | | | |
| Front maturity | 9 | 30 | 42 | 59 | 1 | 8 |
| Other maturities | 12 | 30 | 38 | 61 | 2 | 9 |
| **Manufacturer** | | | | | | |
| Front maturity | 1 | 10 | 11 | 13 | 0 | 3 |
| Other maturities | 22 | 47 | 73 | 89 | 1 | 3 |
| **Producer** | | | | | | |
| Front maturity | 1 | 8 | 11 | 29 | 0 | 1 |
| Other maturities | 17 | 47 | 57 | 87 | 1 | 6 |
| **Floor Broker** | | | | | | |
| Front maturity | 4 | 6 | 10 | 16 | 0 | 0 |
| Other maturities | 20 | 21 | 53 | 52 | 8 | 7 |
| **Swap Dealer** | | | | | | |
| Front maturity | 39 | 6 | 60 | 21 | 2 | 1 |
| Other maturities | 23 | 17 | 44 | 26 | 6 | 2 |
| **Hedge Fund** | | | | | | |
| Front maturity | 23 | 19 | 48 | 50 | 2 | 1 |
| Other maturities | 26 | 28 | 71 | 77 | 7 | 3 |

_Table header (spanning): Crude Oil / Number of Observations: 2964 / (%)_

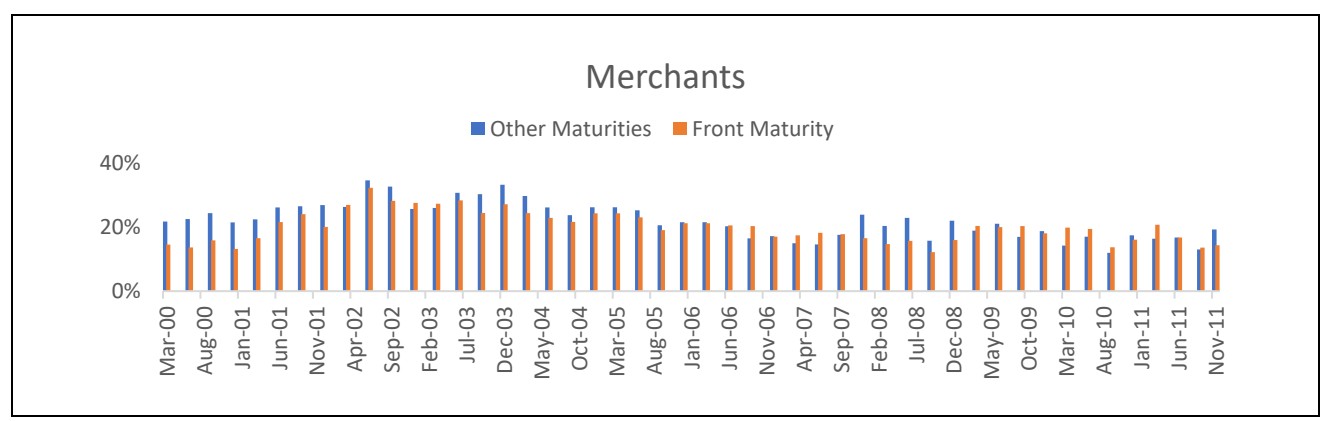

**Figure 2.** *Cont*.

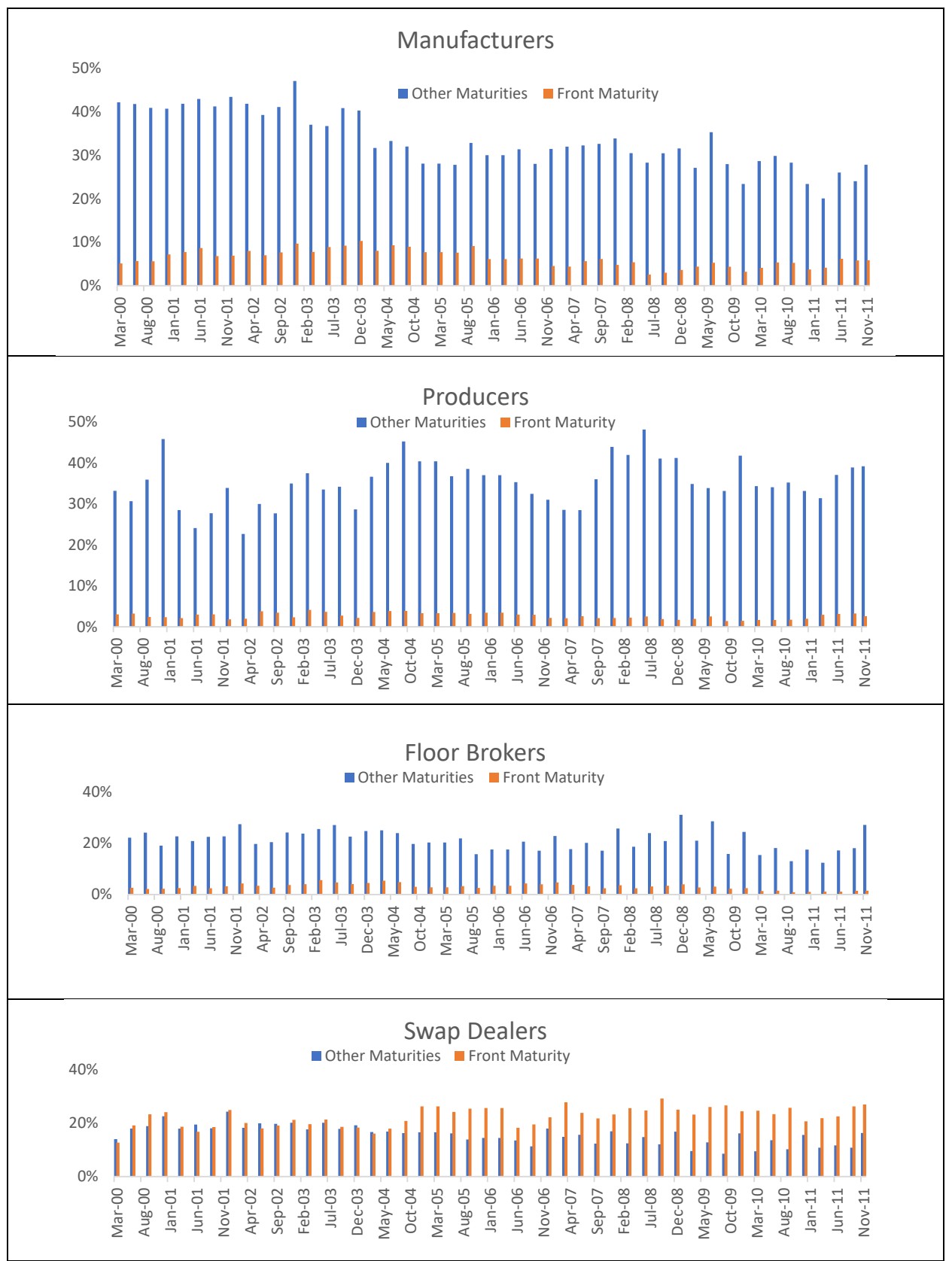

**Figure 2.** *Cont.*

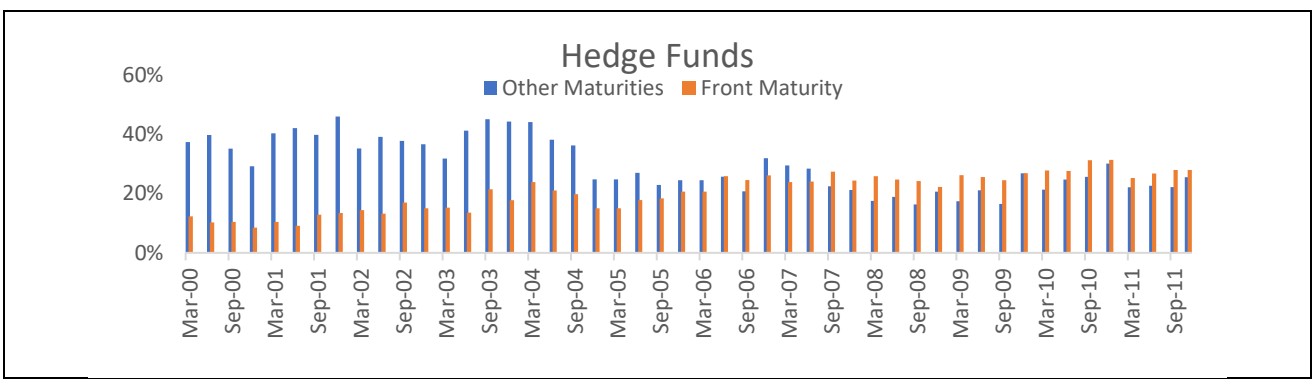

**Figure 2.** Participation rates of institutional investors.

## 4. Market Fundamentals and Institutional Investor Positions

In futures markets, market fundamentals related to the supply and demand of the underlying asset play an important role in the pricing of contracts (e.g., [2,3,10,12]). Fundamentals are also likely to influence trader positioning. To isolate the role of non-fundamental speculative activity from trader positioning due to these factors, we first run regressions of trader position changes on fundamentals. The residuals of these regressions represent trading strategies, market expectations, beliefs, private information, or speculative behavior of the institutional traders we analyze. We use these residuals in the transition probabilities of the model in Section 4 to test whether and how various traders contribute to the probability of low-volatility/bullish or high-volatility/bearish regimes in the crude oil market ([43] labels the dichotomy of bullish/low-volatility and bearish/high-volatility regimes in markets as the "leverage effect." Recent papers examining this asymmetry in volatility include [44–48]). In this setting, fundamentals represent information while "unexplained" positions represent a measure of information processing [49,50] model the dual nature of information and information processing abilities among traders.).

We consider a variety of fundamental variables that affect crude oils prices. First, we consider the [51] (ADS, 2009) business conditions index, designed to track real U.S. business conditions (the ADS index is available from the Federal Reserve Bank of Philadelphia). The ADS index is an accurate measure of the current state of the U.S. real economy. Second, we consider the TED spread—the difference between the interest rates on interbank loans and short-term U.S. government debt. The TED spread is an indicator of perceived credit risk in the economy, with an increase in the TED spread indicating increased U.S. counterparty risk. Third, we consider the MSCI world stock market index, representing the major stock markets around the world (the MSCI World Index is comprised of 1500 world stocks from 24 developed countries and is commonly used as a benchmark for global stock funds. The index is available in local currency and USD, with or without dividends reinvested). While the ADS index and the TED spread mainly refer to the U.S. economy, the MSCI index proxies for global fundamental factors. Fourth, we consider the 10-year expected inflation estimate provided by the Federal Reserve Bank of Cleveland. Fifth, we consider petroleum and other energy liquids inventories published by the U.S. Energy Information Administration since there is substantial evidence that inventories play an important role in the crude oil market (see, e.g., [35,36]. Our inventories exclude the Strategic Petroleum Reserve (SPR) stocks maintained by the Federal Government).

Ref. [33] consider a wide range of macroeconomic announcements and study how crude oil prices react to the unexplained component of these announcements (I unexplained component of the announcements is computed as the difference between the actual announcement and market expectations). For daily data, they find that three announcements have some impact on crude oil prices: the Government Budget Deficit, core CPI, and Housing Starts (the number of privately owned new houses on which construction has been started in a given period). In our analysis, we add these three announcements to capture how fundamentals affect trader positions.

We estimate the following equation, adding a Fourier Transform (of order 5) to account for the possible seasonality in trader positions.

$$\Delta Z_{i,t} = \gamma_0 + \gamma_1 ADS_t + \gamma_2 TED + \gamma_3 \Delta\ln(MSCI_t) + \gamma_4 \Delta EI_t + \gamma_5 Inventories_t + \sum_{k=1}^{3} \gamma_{5+k} Surprise_{k,t}$$
$$+ \sum_{j=1}^{5} \left( a_j sin(2j\pi t) + b_j cos(2j\pi t) \right) + u_{i,t} \tag{1}$$

where $\Delta Z_{i,t}$ is the daily change of institutional invesIor $i$ on day $t$—the change in position represents the daily trading activity of market participants—we standardize position changes so that increases and decreases among position changes are with respect to their averages over the entire sample period; $\Delta\ln(MSCI_t)$ is the daily compounded rate of return of the MSCI index; expected inflation and inventories are non-stationary so we use the first differences; $Surprise_{k,t}$ is the difference between the macro announcement and market expectations for the Government Budget Deficit, core CPI, and Housing Starts (note that in Section 4 we adopt a regime-switching approach which we show clearly provides a better fit for futures returns. For trader positions, we also adopt a non-linear approach and estimate regime-switching models. However, there was no improvement in the fit, and standard information criteria selected the linear model over the non-linear model).

Table 3 reports regression estimates of Equation (1). For the front maturity contract, merchant position changes are positively linked to the TED spread. This implies that when the credit risk in the U.S. economy increases, merchants hedge more. Merchant position changes (in the nearby and other maturities) are also negatively related to the world stock market index. As worldwide economic fundamentals improve, merchants hedge less. In longer maturities, merchant position changes are negatively linked to inventories and positively linked to Housing Starts surprises. When inventories are growing, merchants decrease their hedge positions, perhaps to account for a possible reduction in demand. By the same token, when Housing StIrts increase, the hedging activity of merchants also increases to account for an expected increase in demand for oil.

For the front maturity contract, manufacturer position changes are positively linked to the ADS index—as economic conditions improve, crude oil demand is also expected to grow, and manufacturers increase their hedge positions. Similar to merchants, manufacturers increase their hedging activity when credit conditions deteriorate. In longer maturities, manufacturer position changes are negatively linked to the world fundamentals (as captured by the MSCI index). Interestingly, a positive surprise in CPI reduces hedging activity by manufacturers, perhaps because a positive inflation surprise may indicate that crude oil prices are increasing, hence there is less need for hedging.

For the front maturity contract, producer position changes are positively linked to the ADS economic conditions index and the TED spread. As economic conditions improve, crude oil demand is also expected to grow, and producers increase their hedge positions. Likewise, when credit risk in the U.S. economy increases, producers hedge more. Similar to merchants and manufacturers, producers hedge less when the world stock market index (MSCI) rises. When inventories are growing, producers decrease their hedge positions, perhaps to account for a possible reduction in demand. Positive budget deficit surprises increase producer hedging activity in nearby contracts but not in longer maturities, perhaps because producers are more likely to hedge with real activities rather than with futures.

We find that floor broker activity is unrelated to economic fundamentals, perhaps not surprisingly since these agents take shorter-term positions to intermediate between demand and supply in the pit. However, floor broker positions may be linked to idiosyncratic information in the crude oil market (as we shall see in the next section).

**Table 3.** Trader positions and fundamentals.

| | Merchant | | Manufacturer | | Producer | | Floor Broker | | Swap Dealer | | Hedge Fund | |
|---|---|---|---|---|---|---|---|---|---|---|---|---|
| | Front Maturity | Other Maturities | Front Maturity | Other Maturities | Front Maturity | Other Maturities | Front Maturity | Other Maturities | Front Maturity | Other Maturities | Front Maturity | Other Maturities |
| $\gamma_0$ | −1.5888 (1.7661) | 0.5392 (1.1212) | 4.6657 ** (0.6866) | −0.7371 (0.6347) | 0.4221 (0.3316) | 0.0756 (0.3123) | −0.3867 (0.2492) | 0.2287 (0.2237) | 2.8807 * (1.5227) | −0.4568 (1.5986) | −7.4180 ** (1.8053) | 0.2360 (1.8992) |
| $ADS_t$ | 2.1695 (2.4983) | 0.2343 (0.2157) | 2.4033 ** (0.8340) | 0.1393 (0.6105) | 0.8235 ** (0.3346) | 0.1534 (0.3158) | −0.0039 (0.2680) | −0.2675 (0.2545) | −3.4122 * (1.8956) | −0.1823 (1.4706) | −0.7294 (1.9447) | 0.6031 (1.7988) |
| $TED_t$ | 0.0634 ** (0.03347) | −0.0207 (0.0192) | 0.0210 * (0.0113) | 0.0144 (0.0103) | 0.0160 ** (0.0057) | −0.0020 (0.0052) | 0.0037 (0.0044) | −0.0070 (0.0050) | −0.0544 * (0.0300) | 0.0053 (0.0272) | −0.0341 (0.0272) | 0.0160 (0.0300) |
| $\Delta\ln(MSCI_t)$ | −2.1548 ** (0.9698) | −3.8067 ** (0.6987) | −0.3751 (0.3827) | −1.2568 ** (0.4059) | −0.6423 ** (0.2054) | −0.4133 ** (0.1979) | 0.2019 (0.1941) | 0.6487 ** (0.2731) | 2.3068 ** (0.9280) | −0.0384 (1.1537) | 5.2857 ** (1.2623) | 8.5744 ** (1.8206) |
| $\Delta EI_t$ | −4.6540 (3.3164) | −3.6727 (34788) | 0.5378 (1.3860) | −0.1701 (1.5225) | −0.1879 (0.6713) | −0.4466 (0.7529) | −0.6926 (0.6812) | −0.2894 (0.8005) | 1.2667 (2.8192) | −5.2782 (3.6666) | 6.5001 (5.2041) | 10.376 * (5.9575) |
| $Inventories_t$ | −0.0077 (0.0350) | −0.0037 * (0.0021) | −0.0085 0.0146 | 0.0056 (0.0119) | −0.0148 ** (0.0071) | 0.0024 (0.0067) | 0.0044 (0.0053) | 0.0047 (0.0042) | 0.0885 ** (0.0295) | 0.0215 (0.0258) | −0.0681 ** (0.0337) | −0.0549 * (0.0325) |
| Budget Deficit | 3.8560 (2.6652) | 5170.6 (3479.1) | −0.6174 (0.7922) | 0.8336 1.1505 | 1.4622 ** (0.7697) | 0.4978 (0.7486) | −0.3244 (0.6616) | −1.0597 (0.7613) | −2.2774 * (1.1669) | −0.8638 (3.1833) | −5.7148 ** (2.7570) | −5.2473 (3.8557) |
| Core CPI | 5.8412 (5.1973) | 1.6588 (4.3930) | −1.5250 (1.9788) | −4.8048 ** (2.2581) | 0.6086 (1.130) | 0.5875 (1.7493) | 0.4792 (1.6523) | 0.3033 (2.3796) | −6.2108 * (3.7167) | −1.2521 (4.7672) | −2.4538 (5.5578) | 7.4428 (5.9089) |
| Housing Starts | 0.0041 (0.0048) | 0.0086 * (0.0050) | −0.0022 (0.0022) | −0.0031 (0.0027) | −0.0003 (0.0012) | −0.0028 (0.0018) | −0.0027 (0.0018) | −0.0026 (0.0023) | −0.0003 (0.0039) | −0.0051 (0.0065) | 0.0053 (0.0052) | 0.0068 (0.0052) |
| $R^2$ (%) | 24.2 | 22.8 | 22.9 | 27.2 | 12.1 | 6.03 | 2.88 | 2.54 | 50.5 | 7.82 | 2.78 | 2.99 |

\*\* and \* indicate 5 percent and 10 percent significance levels, respectively.

As economic conditions improve, swap dealers reduce long positions in the crude oil market. Changes in front-month swap dealer position are linked to most fundamentals considered in Equation (1). In the front maturity, fundamentals explain more than 50 percent of the variation in swap dealer positions. Consistent with swap dealers bringing index investment primarily to the nearby contract, their position changes in other maturities are unrelated to fundamentals.

Table 3 also shows that while hedge fund position changes are linked to fundamentals, fundamentals explain very little of the variation in hedge fund activity. When the world stock market improves, hedge fund activity in crude oil increases, while increases in inventories reduce hedge fund trading activity.

Overall, hedger trading activity is strongly linked to demand effects (as captured by the ADS index and the TED spread) while speculative positions from hedge funds are linked to inventories and world economic conditions. Interestingly, front-month swap dealer positions are heavily linked to fundamentals, but swap dealer activity in more distant contracts is largely invariant to fundamentals.

## 5. Regime-Switching Modeling

Figure 1 shows several dynamics in the price of crude oil. In particular, note that price downturns are associated with higher volatility (bear markets are typically associated with higher volatility levels [28,52]. Ref. [15] demonstrate the strong link between bear markets and volatility for five futures markets, including crude oil). This behavior is evident in September–November 2001 (when prices fell to their lowest levels in nearly two years amid fears of a recession), in March 1993 (when military action began in Iraq), and in October-December 2004 (when Russia ratified the Kyoto Protocol and OPEC agreed to cut production to official quota levels). At the same time, when prices increase, volatility is lower.

Our regime switching approach captures these features of the data. We distinguish between two different regimes in financial markets—increasing and decreasing prices—which exhibit different (unconditional) mean returns and different return variances. Econometrically, our GARCH model captures the leptokurtosis, volatility clustering, and heteroskedasticity present in returns as well.

### 5.1. The Model

Refs. [52,53] introduce Markov regime switching in the GARCH framework

$$y_t = \mu(S_t) + \varepsilon_t$$

$$\varepsilon_t = \sigma_t u_t \qquad u_t \sim \text{i.i.d.} N(0,1) \tag{2}$$

$$\sigma_t^2(S_t, S_{t-1}, \dots, S_0) = \omega(S_t) + \sum_j^p \alpha_j(S_{t-j}) \varepsilon_{t-j}^2 + \sum_j^q \beta_j(S_{t-j}) \sigma_{t-j}^2(S_{t-1}, \dots, S_0).$$

where $y_t$ represent returns at time $t$ for crude oil. The innovation term, $\varepsilon_t$, is normally distributed (for stock market data, [53] adopts a Student t-distribution for the error term. When we assume a Student t-distribution for the innovations, the estimated degrees of freedom are very high, so we adopt a normal distribution) and the constant in the conditional mean equation, $\mu$, is allowed to switch between two regimes—positive mean ($\mu_1$), which is accompanied by relatively low volatility, and negative mean ($\mu_0$), which is accompanied by relatively high volatility, so that

$$\mu(S_t) = \mu_1 S_t + \mu_0(1 - S_t)$$

$$S_t \in \{0, 1\} \ \forall t$$

$$Pr(S_t = 0 | S_{t-1} = 0) = p_{00}$$

$$Pr(S_t = 1 | S_{t-1} = 1) = p_{11.}$$

The volatility literature demonstrates that the GARCH(1,1) model is able to capture the volatility dynamics of asset returns, e.g., [54,55]. Our data confirm this result for the crude oil futures markets. Therefore, in what follows we concentrate on the GARCH(1,1) case. The conditional variance, $\sigma_t^2$, in Equation (2) is a function of the entire history of the state variable $S_t$. This is due to the autoregressive term, $\sigma_{t-j}^2$, in the conditional variance equation (see [28,53,56]). Following [53], we approximate the entire history using only the two most recent values of the state variable, a procedure that is parsimonious with evaluating the likelihood function and makes the conditional variance, $\sigma_t^2$, a function of only the current ($S_t$) and the previous states ($S_{t-1}$). By integrating out $S_{t-1}$, the conditional variance for the GARCH(1,1) can therefore be written as

$$\sigma_t^2(a,b) = \omega(S_t = a) + \alpha\left[\varepsilon_{t-1}^2(S_{t-1} = b)\right] + \beta\sigma_{t-1}^2(S_{t-1} = b) \tag{3}$$

Equation (3) allows the constant in the conditional variance equation to switch, which, in turn, allows the unconditional variances to switch across regimes (following [53], $\omega(S_t)$ is parameterized as $\gamma(S_t)\cdot\omega$ such that $\gamma(S_t = 1)$ is normalized to unity. Similarly, [57] use a regime-switching approach to detect bubbles in oil price).

In this basic setup the transition probabilities are constant, which we deem overly restrictive. Indeed, we are interested specifically in whether transition probabilities depend on the trading activity of market participants, so we introduce time varying probabilities modeled as probit functions of unexplained trader positions from our fundamental macroeconomic regressions above, denoted by $u_{i,t}$

$$Pr(S_t = 0|S_{t-1} = 0, u_{i,t-d}) = p_{00,t} = \Phi\left(u'_{i,t-d}\varsigma\right) \tag{4}$$

$$Pr(S_t = 1|S_{t-1} = 1, u_{i,t-d}) = p_{11,t} = \Phi\left(u'_{i,t-d}\upsilon\right). \tag{5}$$

Here $\Phi$ denotes the cumulative density function of the normal distribution, and $\varsigma$ and $\upsilon$ are parameters that capture how the transition probabilities vary in response to investor positioning. This approach estimates the conditional probability of being in a given regime at time $t$ given the information available at time $t - 1$. We test the temporal dimension of [30] by allowing $d$ to vary from 1 to 5—i.e., from the previous day ($d = 1$) to the previous week ($d = 5$).

Denoted by $\hat{\xi}_{t|t}$, the ($N \times 1$) vector of conditional probabilities and defines $\eta_t$ as the ($N \times 1$) vector of the conditional density of returns $y_t$ conditional on $S_t$ and $S_{t-1}$. (Given that the Markov process has 2 states, $N = 4$). Following [58], the optimal forecast at each time $t$ is computed by iterating the following equations

$$\hat{\xi}_{t|t} = \frac{\left(\hat{\xi}_{t|t-1} \odot \eta_t\right)}{1'\left(\hat{\xi}_{t|t-1} \odot \eta_t\right)}$$

$$\hat{\xi}_{t|t+1} = P_{t+1} \cdot \hat{\xi}_{t|t}$$

where $\mathbf{1}$ denotes the unit vector, $P_{t+1}$ is the ($N \times N$) Markov transition probability matrix, and $\odot$ denotes element-by-element multiplication. In this framework, $P_{t+1}$ is time varying as a function of market participant positions. This approach allows us to compute the probability of moving from one regime to the other (and vice versa) in period $t + 1$ given the trading behavior of market participants at time $t$.

We estimate the parameters by maximizing the following likelihood function

$$ln[L_t(a,b)] = -\frac{1}{2}ln\left[\sigma_t^2(a)\right] - \left[\frac{\varepsilon_{t-1}^2(a)}{2\sigma_t^2(b)}\right] - ln\left[\sqrt{2\pi}\right] \tag{6}$$

where $a \in \{0,1\}$ relates to $S_t \in \{0,1\}$ and $b \in \{0,1\}$ relates to $S_{t-1} \in \{0,1\}$ (see Hamilton, 1994).

*5.2. Parameter Estimates*

We utilize detailed position data to estimate the model in Equation (6). Importantly, we use the residuals from Equation (1) to capture private information and/or trading strategies of institutional traders. Indeed, we are interested specifically in whether price continuations and reversals (the transition probabilities) depend on trading activity (unrelated to fundamentals) from hedge funds, swap dealers, and hedgers.

Table 4 presents the results. The first two columns refer to the model with constant transition probabilities. We benchmark to a simple GARCH(1,1) model with no regime switching (not reported) which generates log-likelihoods of $-7502$ and $-6638$ for the front maturity and longer maturity returns, respectively. Each of our log-likelihoods is substantially higher, indicating that our regime switching approach provides a much better fit for modeling crude oil returns and volatility.

The first two columns in Table 4 indicate that regime 0 is characterized by positive returns (i.e., increasing prices), while the regime 1 is characterized by negative returns (i.e., decreasing prices). The volatility in regime 1, when prices are decreasing, is about 11 and 9 times larger ($\gamma$) with respect to regime 0 for the front maturity and longer maturities, respectively. This is in line with previous research in stock markets (e.g., [59,60]). On average, daily price variations are much smaller during upward price movements (regime 0), suggesting that the crude oil market goes up more smoothly than it goes down. As demonstrated by the parameter estimates of $\alpha$ and $\beta$, the volatility process is persistent but stationary ($\alpha + \beta < 1$).

In the rest of the table, we allow the transition probabilities to be time varying as a function of unexplained (by fundamentals) institutional trading—the residuals from Equation (1). Our analysis effectively incorporates trader activity indirectly in the conditional probability of moving between the two regimes (our prior filtering of trader positions by fundamentals as in Equation (1) and allowing for 1- to 5-day lags in the transition probabilities should resolve any possible endogeneity issue). The log-likelihood notably improves when we allow the transition probabilities to be time varying.

Merchant trading activity in longer maturities increases the probability of remaining in the high-volatility regime with decreasing prices—merchants significantly enter transition probability $P_{11}$. The probability of remaining in either regime increases with manufacturer trading activity in longer maturities—manufacturers have a positive and significant coefficient in both $P_{00}$ and $P_{11}$. Producer activity increases the probability of remaining in the low-volatility, increasing price regime (producer coefficients are significantly positive in $P_{00}$).

Hedgers as a group (merchants, manufacturers, and producers) exhibit similar results in terms of the signs and magnitudes of our estimated coefficients and in terms of goodness of fit (higher log-likelihood values). Overall, we find that hedging activity increases the probability that crude oil prices (and volatility) stay in the same regime but hedging activity does not contribute to price reversals.

Floor broker positions in the front month contract also increase the probability of the crude oil market remaining in a regime with a significantly positive coefficient in both transition probabilities $P_{00}$ and $P_{11}$. Conversely, swap dealers are not significantly related to the transition probabilities—their non-fundamentally driven positions bring no incremental information to our model. In fact, the log-likelihood values for swap dealers are the lowest among all institutional traders.

Hedge fund positions, however, significantly decrease the likelihood of remaining in the same regime, which is evidence that hedge funds largely serve to stabilize futures markets by positioning against oil market price trends (similar evidence is found in [15,61]). Importantly, the value of the log-likelihood is the highest for hedge funds, indicating that they bring relatively more information about price reversals than other traders do.

**Table 4.** Regime Switching Estimates.

| | Constant Transition Probabilities | | Merchant | | Manufacturer | | Producer | |
|---|---|---|---|---|---|---|---|---|
| | Front Maturity | Other Maturities | Front Maturity | Other Maturities | Front Maturity | Other Maturities | Front Maturity | Other Maturities |
| $\mu_0$ | 0.1532 ** | 0.1945 ** | 0.1352 ** | 0.1546 ** | 0.1381 ** | 0.1587 ** | 0.1311 ** | 0.1589 ** |
| | (0.0498) | (0.0433) | (0.0483) | (0.0349) | (0.0394) | (0.0346) | (0.0576) | (0.0336) |
| $\mu_1$ | −1.1376 ** | −0.8639 ** | −1.1376 * | −0.7996 ** | −1.1909 ** | −0.7991 ** | −1.1971 ** | −0.9019 ** |
| | (0.5206) | (0.4192) | (0.6103) | (0.3340) | (0.3438) | (0.2586) | (0.4095) | (0.2897) |
| $\Omega$ | 0.2187 ** | 0.1875 ** | 0.1781 * | 0.1368 ** | 0.1766 ** | 0.1666 ** | 0.1813 ** | 0.1110 ** |
| | (0.0941) | (0.6781) | (0.1042) | (0.0550) | (0.0515) | (0.0688) | (0.0584) | (0.0365) |
| $\gamma$ | 10.963 ** | 9.1689 ** | 10.397 * | 9.5917 ** | 10.449 ** | 9.2618 ** | 9.6591 ** | 10.374 ** |
| | (4.6685) | (3.0188) | (5.1668) | (2.5095) | (2.1864) | (2.1424) | (2.3853) | (2.6853) |
| $\alpha$ | 0.0182 ** | 0.0273 ** | 0.0130 ** | 0.0237 ** | 0.0293 ** | 0.0160 ** | 0.0191 ** | 0.0307 ** |
| | (0.0068) | (0.0138) | (0.0034) | (0.0079) | (0.0100) | (0.0063) | (0.0026) | (0.0128) |
| $\beta$ | 0.9320 ** | 0.9435 ** | 0.9320 ** | 0.9235 ** | 0.9324 ** | 0.9101 ** | 0.9291 ** | 0.9347 ** |
| | (0.0298) | (0.0269) | (0.0177) | (0.0260) | (0.0165) | (0.0262) | (0.0201) | (0.0170) |
| $P_{00}$-Const | 2.6618 ** | 2.7560 ** | 2.1701 ** | 2.0675 ** | 2.1816 ** | 2.0748 ** | 2.2462 ** | 2.2400 ** |
| | (0.3144) | (0.4164) | (0.1423) | (0.2136) | (0.1235) | (0.1497) | (0.2367) | (0.1448) |
| $P_{00}$-$u_{i,t-1}$ | | | 0.1703 | −0.0067 | −0.1154 | 0.2321 * | 0.0274 | 0.3793 ** |
| | | | (0.4039) | (0.0295) | (0.1157) | (0.1374) | (0.1408) | (0.1144) |
| $P_{11}$-Const | 1.066 ** | 1.5173 ** | 1.0968 ** | 1.1524 ** | 1.1395 ** | 1.0860 ** | 1.1885 ** | 1.0625 ** |
| | (0.3821) | (0.3516) | (0.3049) | (0.2352) | (0.1724) | (0.1893) | (0.2628) | (0.1691) |
| $P_{11}$-$u_{i,t-1}$ | | | −0.2220 | 0.3778 ** | −0.3484 | 0.3540 ** | −0.1774 | −0.1138 |
| | | | (0.7419) | (0.1249) | (0.2291) | (0.1746) | (0.3422) | (0.2487) |
| $\theta$ | −0.0386 ** | −0.0668 ** | −0.0381 * | −0.0629 ** | −0.0389 * | −0.0630 ** | −0.0369 * | −0.0654 ** |
| | (0.0189) | (0.0258) | (0.0200) | (0.0189) | (0.0235) | (0.0187) | (0.0224) | (0.0193) |
| Log-Lik. | −6523.3 | −6022.9 | −6510.8 | −6008.6 | −6510.9 | −6008.5 | −6512.7 | −6008.4 |

| | Floor Broker | | Swap Dealer | | Hedge Fund | |
|---|---|---|---|---|---|---|
| | Front Maturity | Other Maturities | Front Maturity | Other Maturities | Front Maturity | Other Maturities |
| $\mu_0$ | 0.1295 ** | 0.1501 ** | 0.1313 ** | 0.1545 ** | 0.1428 ** | 0.1583 ** |
| | (0.0427) | (0.0342) | (0.0430) | (0.0482) | (0.0406) | (0.0351) |
| $\mu_1$ | −1.1570 ** | −0.7606 ** | −1.1347 ** | −0.8399 ** | −1.2799 ** | −0.8146 ** |
| | (0.4573) | (0.2380) | (0.4027) | (0.3051) | (0.3607) | (0.2408) |

**Table 4.** *Cont.*

| | Constant Transition Probabilities | | Merchant | | Manufacturer | | Producer | |
|---|---|---|---|---|---|---|---|---|
| | Front Maturity | Other Maturities | Front Maturity | Other Maturities | Front Maturity | Other Maturities | Front Maturity | Other Maturities |
| $\Omega$ | 0.1868 ** | 0.1579 ** | 0.1933 ** | 0.1395 ** | 0.1457 ** | 0.1228 ** | | |
| | (0.0555) | (0.0536) | (0.0572) | (0.0639) | (0.0524) | (0.0508) | | |
| $\gamma$ | 9.6957 ** | 8.9099 ** | 9.5823 ** | 8.9379 ** | 11.628 ** | 10.670 ** | | |
| | (2.0599) | (1.9243) | (2.0111) | (2.7250) | (3.0425) | (3.0146) | | |
| $\alpha$ | 0.0253 ** | 0.0178 ** | 0.0294 * | 0.0285 ** | 0.0358 ** | 0.0255 ** | | |
| | (0.0108) | (0.0072) | (0.0176) | (0.0105) | (0.0172) | (0.0130) | | |
| $\beta$ | 0.9300 ** | 0.9171 ** | 0.9280 ** | 0.9204 ** | 0.9400 ** | 0.9301 ** | | |
| | (0.0173) | (0.0205) | (0.0170) | (0.0180) | (0.0221) | (0.0205) | | |
| $P_{00}$-Const | 2.2443 ** | 2.0985 ** | 2.2164 ** | 2.1706 ** | 2.1714 ** | 2.0227 ** | | |
| | (0.1369) | (0.1185) | (0.1613) | (0.1510) | (0.1691) | (0.1151) | | |
| $P_{00}$-$u_{i,t-1}$ | 0.1258 ** | −0.0214 | −0.0673 | 0.1805 | −0.1770 * | −0.1329 ** | | |
| | (0.0605) | (0.0345) | (0.0975) | (0.2031) | (0.1071) | (0.0412) | | |
| $P_{11}$-Const | 1.1670 ** | 1.1002 ** | 1.1502 ** | 1.1548 ** | 1.2168 ** | 1.2232 ** | | |
| | (0.1885) | (0.1781) | (0.1964) | (0.2522) | (0.2110) | (0.2045) | | |
| $P_{11}$-$u_{i,t-1}$ | 0.1304 ** | −0.0339 | 0.0356 | −0.1028 | −0.6417 ** | −0.4775 ** | | |
| | (0.0514) | (0.0283) | (0.0669) | (0.3229) | (0.2006) | (0.1446) | | |
| $\theta$ | −0.0375 * | −0.0622 ** | −0.0379 ** | −0.0624 ** | −0.0410 ** | −0.0626 ** | | |
| | (0.0215) | (0.0194) | (0.0187) | (0.0229) | (0.0184) | (0.0194) | | |
| Log-Lik. | −6509.8 | −6010.2 | −6515.2 | −6011.3 | −6508.9 | −6006.5 | | |

** and * indicate 5 percent and 10 percent significance levels, respectively.

The results in Table 4 consider trading activity of institutional traders at $t - 1$—the previous day. We also test activity for days $t - 2$ until $t - 5$ and find that the effects of trading activities on price reversals fade away over longer lags (likelihood-based tests show that previous day trading activity produces the best fit). Our results suggest that Abreu and Brunnermeier's [30] concept of "*delayed arbitrage*" is on the order of a few days and confirm that synchronization risk in oil markets is present only in the very short run.

*5.3. Transition Probabilities: A Forecasting Exercise*

We next explore whether the transition probabilities, conditional on trader behavior, help forecast regime switches. To do so, we estimate Equation (6) from January 2000 to December 2003, a period of relative oil price stability. We then compute and store the transition probabilities for the first out-of-sample day using Equations (4) and (5). We do this recursively, adding each subsequent trading day to the estimation window through the end of 2011.

Figure 3 displays the price of crude oil along with the time series of transition probability estimates—we report the 44-day moving average of the transition probabilities. We concentrate on $p_{10,t}$, which represents the probability of switching from the falling price/high-volatility state to the rising price/low-volatility state.

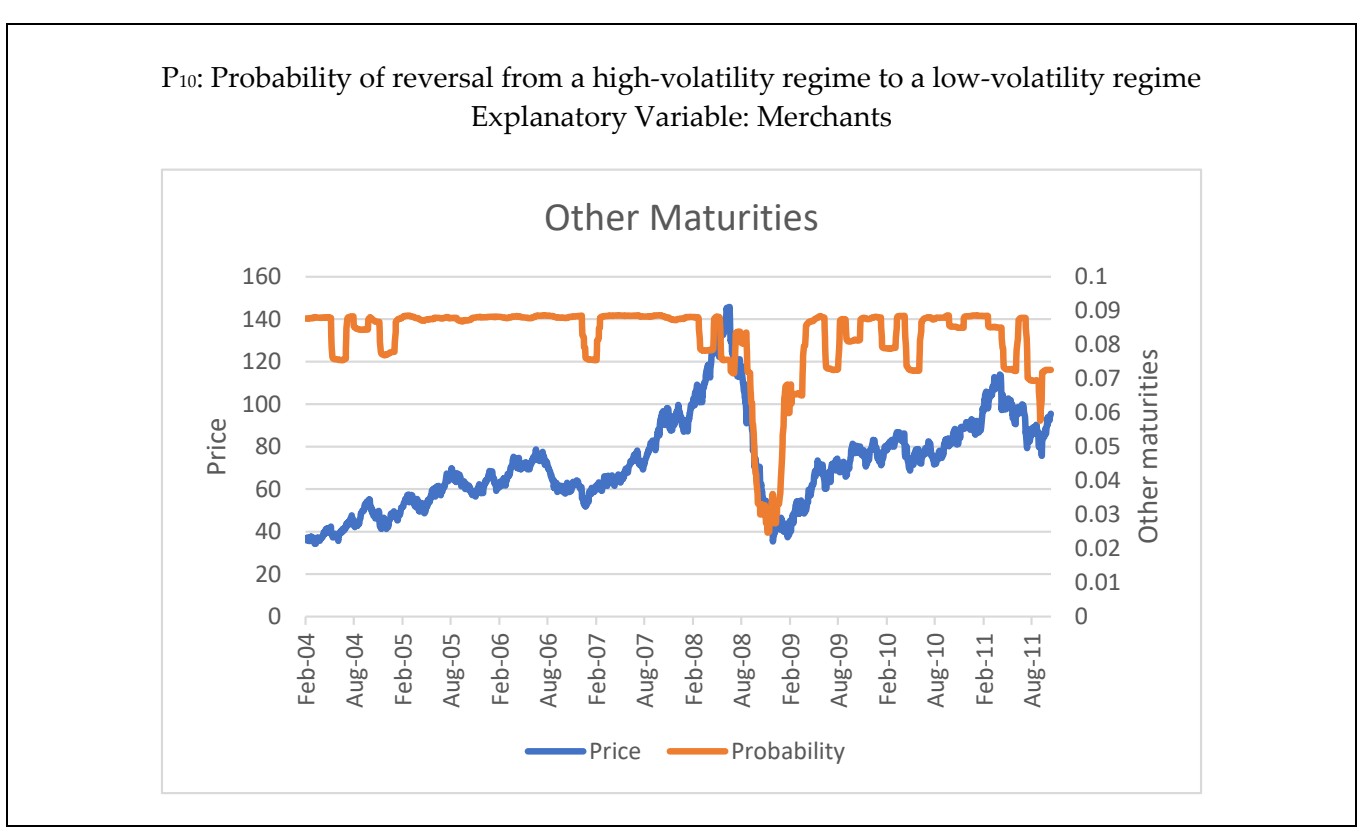

**Figure 3.** *Cont.*

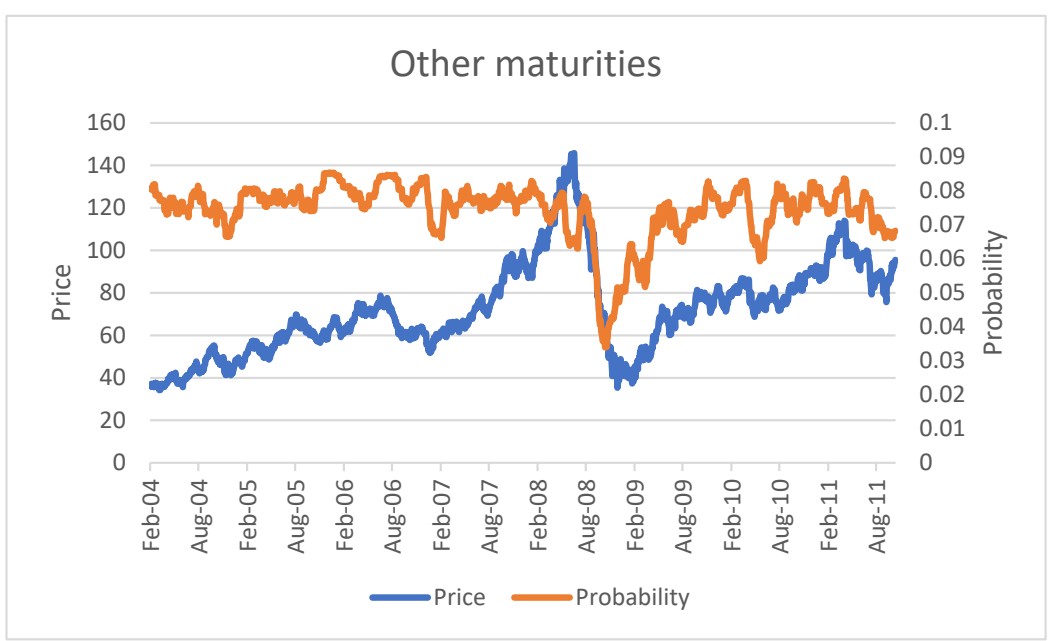

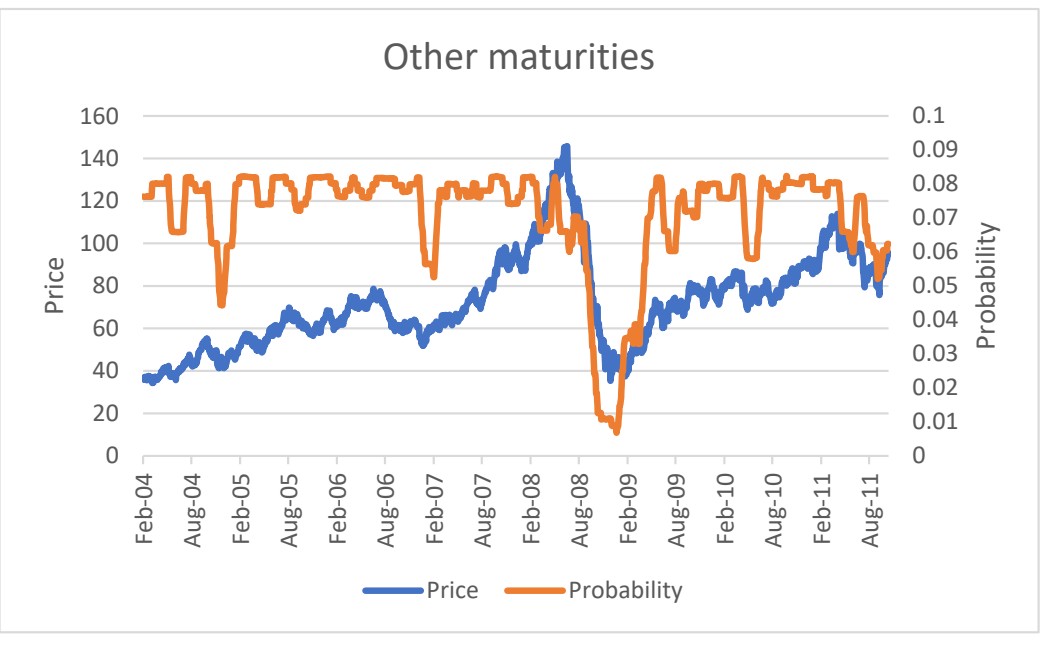

**Figure 3.** *Cont.*

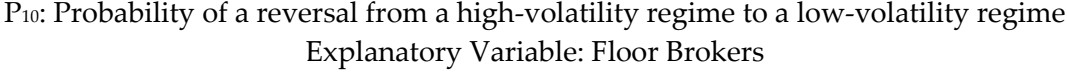

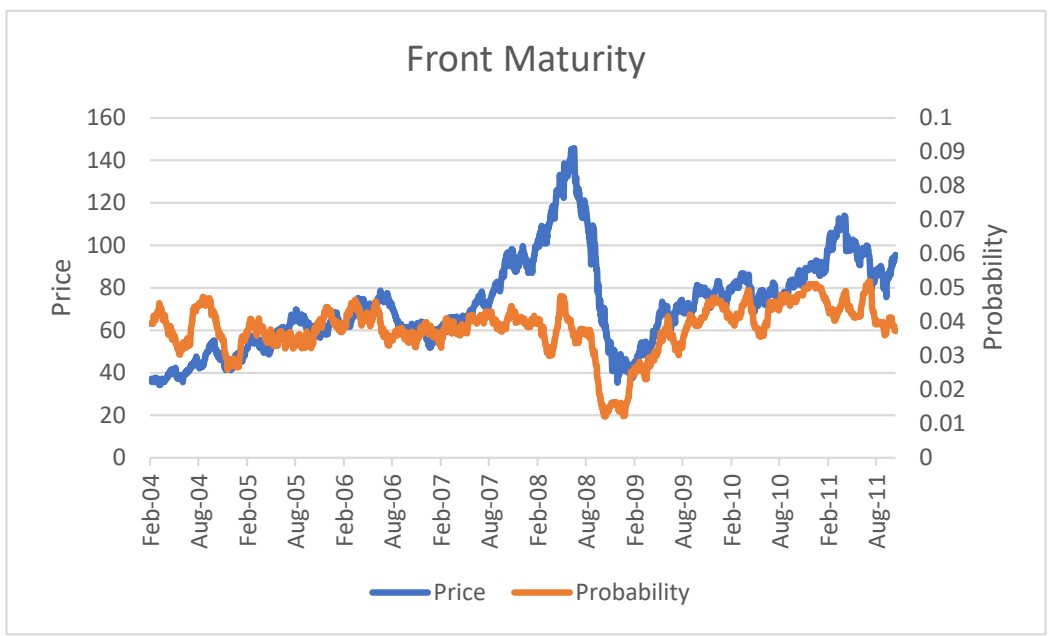

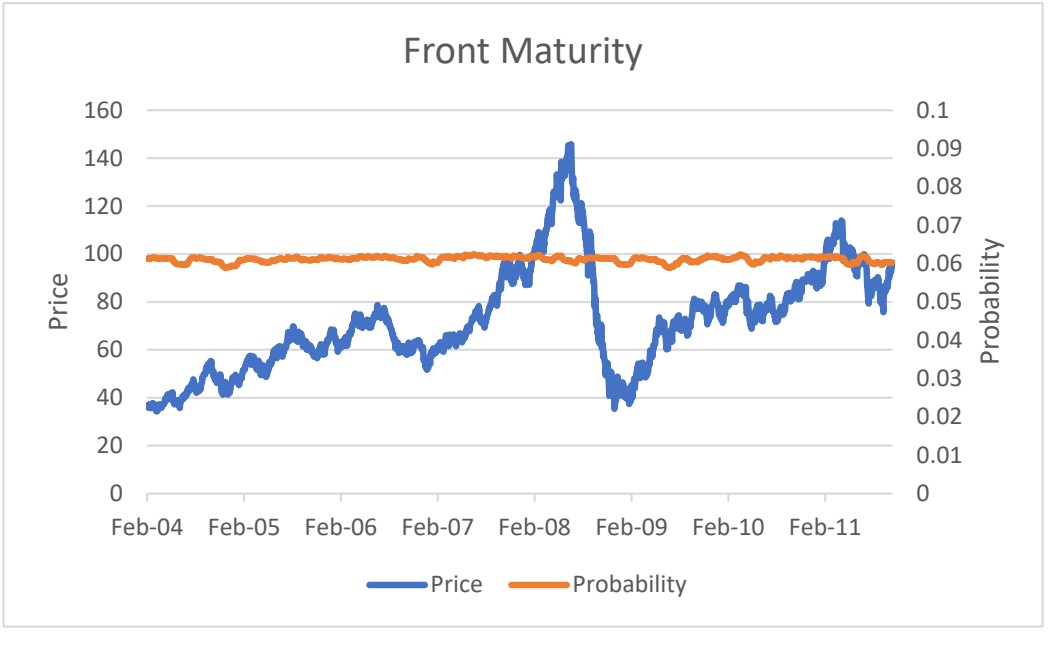

**Figure 3.** *Cont.*

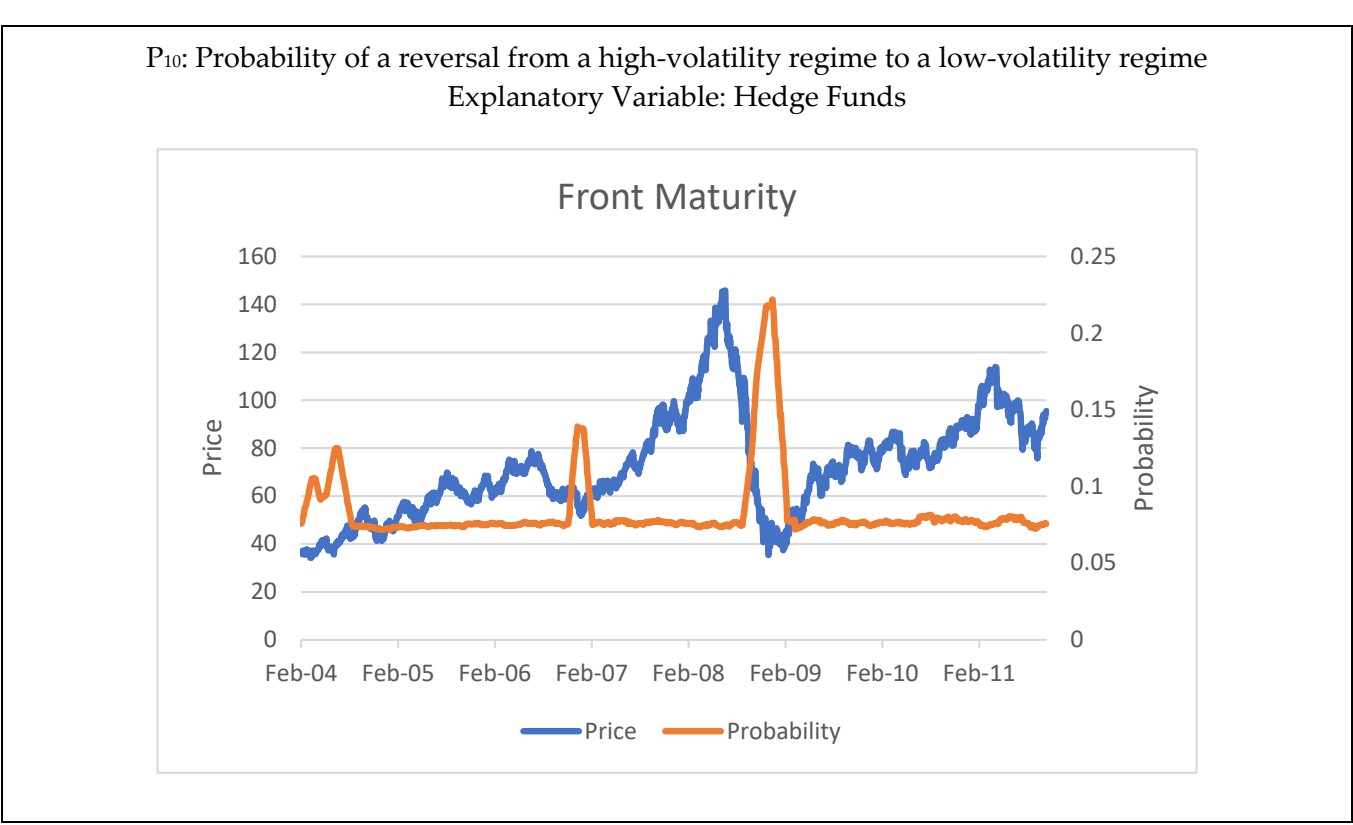

**Figure 3.** Crude Oil Prices and Estimated Transition Probabilities.

The probability of moving from a turbulent state to a more stable state conditional on merchant trading behavior averages just 10 percent and decreases considerably during the financial crisis (for merchants, manufacturers, and producers, we only report results for longer maturities. Results from the front maturity contract are in line with what we report here but weaker). A similar result is evident for manufacturers and producers. These hedger groups reduce the probability of moving to a less volatile state, or alternatively, increase the probability of remaining in the same state ($p_{10,t} = 1 - p_{11,t}$), as shown in Table 4.

Floor broker trading activity also reduces the probability of moving from a falling price/high-volatility regime to a rising price/low-volatility regime. We only report results for the front maturity contract, since the transition probability for longer maturities is largely unrelated to floor broker positions.

Hedge funds are the only institutional traders that increase the probability of moving from falling price/high-volatility regimes to rising price/low-volatility regimes. Notably, in January 2009, this probability jumped from 7 percent to 22 percent—an indication that the conditional probability (based on hedge fund positions) of a crude oil price reversal increased substantially. Note that the transition probabilities we estimate only indicate the conditional probability of moving from one regime to another based on recent hedge fund positioning and it would be incorrect to interpret these results as evidence that hedge funds move the market. Rather, these results suggest that aggregate hedge fund position changes reflect information processing by hedge funds above and beyond fundamental market information that is useful in estimating the probability of future price reversals. Similar results apply to the probability of remaining in the same state ($p_{10,t}$) for longer maturities (we do not report results for $p_{01,t}$, the probability of moving from a low-volatility regime to a high-volatility regime. These results are not as clear cut as those for $p_{10,t}$).

## 6. Conclusions

In recent years, many assets, including crude oil, experienced sustained price increases followed by sudden price decreases. At the same time, hedge funds, swap dealers, and arbitrageurs dramatically increased their activity in these markets. We first analyze the relations between trader positions and fundamentals and find significant evidence that fundamentals drive trader positions in some manner.

In the spirit of Abreu and Brunnermeier [30,32] where synchronization risk affects price patterns, we exploit the confluence of trader positioning conditional on fundamental factors to explore whether institutional trades are useful to predict continuations or reversals of price trends in crude oil markets. In fact, the contribution of our work is to examine the role of institutional traders in switching from a high-volatility regime to a low-volatility regime and vice versa. We propose and estimate a Markov switching model between rising price/low-volatility and falling price/high-volatility regimes conditioned on the unexplained (by fundamentals) positions of institutional traders.

We find hedge fund activities add incrementally to the transition probabilities across regimes, suggesting that information processing by hedge funds contributes significantly to the probability of continuations and reversals in oil markets. Our evidence is consistent with synchronization behavior among market arbitrageurs, as modeled by Abreu and Brunnermeier [30,32].

Conversely, we find swap dealer positions are largely unrelated to transition probabilities between bull and bear markets, consistent with their relatively benign diversification goal of gaining long exposure to oil markets [15]. Hedgers, on the other hand, facilitate the persistence in price/volatility regimes and do not signal regime changes.

To the best of our knowledge, this is the first study using regime switching models to analyze how and whether institutional traders contribute to regime changes. Our results demonstrate that trader positioning can be useful in predicting the transition probability of moving between falling price/high-volatility and rising price/low-volatility oil market regimes. Our Markov switching approach represents an important step toward a better understanding of the determinants of price patterns in crude oil. Although not directly addressing the existence or causes of asset bubbles, we find that institutional positions are informative (in the conditional sense) about the transition probabilities between oil market regimes.

**Author Contributions:** Conceptualization all authors; methodology all authors; software, all authors; validation, all authors; formal analysis, all authors; investigation, all authors; resources, all authors; data curation, all authors; writing—original draft preparation, all authors; writing—review and editing, all authors; visualization, all authors; supervision, all authors; project administration, all authors. All authors have read and agreed to the published version of the manuscript.

**Funding:** This research received no external funding.

**Informed Consent Statement:** Not applicable.

**Data Availability Statement:** The datasets presented in this article are not readily available because they contain confidential supervisory information and cannot be shared with the general public.

**Conflicts of Interest:** The authors declare no conflict of interest.

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
