# Peer review of "Crude Oil Price Movements and Institutional Traders"

_commodities, doi:10.3390/commodities3010006_

Round 1

Reviewer 1 Report

Comments and Suggestions for Authors

The paper Crude Oil Price Movements and Institutional Traders covers a very interesting topic. It has a correct and traditional patern - Introductio / background, lietarture review, methods, results and discussion and conclusion - or sligtly modified. all sections with enough information. The literature review is very comprehensive and up-to-date. the quality of the english language is good. Thus, in my opinion, the paper has the quality needed to be published as it is.

Author Response

Thank you for your comments. Based on other feedback we have made minor changes to the text and added a sentence to the abstract.

Reviewer 2 Report

Comments and Suggestions for Authors

This article examines the role which hedge fund, swap dealer, and arbitrageur activity, respectively, has in the crude oil market.  For the empirical analysis, it draws on position data from institutional investors in order to analyze linkages between the traders' positions and fundamentals. The major finding of this article is that institutional position changes reflect economic factors. The Markov regime-switching framework is applied and, using time varying probabilities, finds that institutional traders' position changes contribute incrementally to regime change probabilities.

Overall, this is an excellent and well-executed article. Its findings are topical for both academics and practitioners.  The discussion on "delayed arbitrage" is particularly interesting and will make for good future discussion in the field.

Given the aforementioned, I recommend an "Accept" on this one.

Some additional comments I have are as follows:

(a) This article uses a variety of fundamental variables are considered which can affect crude oil prices. For example, incorporating the Aruoba-Diebold-Scotti (ADS) business conditions index into the analysis is very interesting, as well as the TED spread and MSCI world index.

(b) The Fourier transformation in Eq. (1) is a good approach in order to deal with possible seasonalities in trader positions. Such seasonalities are often not addressed in commodity futures research.

(c) This article quantifies (or, gives some good guidance for) the "delayed arbitrage" concept of Abreu and Brunnermeier (2002). This is an interesting finding.

Author Response

(The authors gave the same response as above.)

Reviewer 3 Report

Comments and Suggestions for Authors

The authors use Markov RS modelling to examine the role of institutional positions in regime changes. The study in general is well-written and has interesting data. The model used is established. The results are not robust, but provide the opportunity to draw conclusions.

The weakest part of the study that must be significantly amended is the introduction.

The following changes must be made

1. Terminology. Authors must replace , the term 'works' (line 24-26, page 1) by the term 'studies'.

2. Rewriting of phrases. The phrases line 23-33, page 1 is not connected with what follows. I suggest the authors add the phrase

a. 'In this study', in the beginning of line 34.

b. 'This is in line to other studies' or a similar phrase on line 41, page 2

3. Authors should move line 93-114 text to line 40.

4. Text of line 41-93, has to be rewritten and form a different Section that should be named '2. Literature Review and Discussion'. Because this new Section will be added, the remaining sections must be renumbered to Section 3. Data etc....

5. Abstract and Conclusion have to be rewritten so to emphasize on the contribution of the study ie ' to examine the role of institutional positions in regime changes.

Apart from the above, I dont recvommend any other changes, and I am happy the paper to be accepted for publication after these minor amendmends are made.

Comments on the Quality of English Language

-

Author Response

The following changes must be made:

  1. Authors must replace the term 'works' (line 24-26, page 1) by the term 'studies'. Done.

2. Rewriting of phrases. The phrases line 23-33, page 1 is not connected with what follows. I suggest the authors add the phrase

a. 'In this study', in the beginning of line 34. Done.

b. 'This is in line to other studies' or a similar phrase on line 41, page 2. Done.

3. Authors should move line 93-114 text to line 40. Done.

4. Text of line 41-93, has to be rewritten and form a different Section that should be named '2. Literature Review and Discussion'. Because this new Section will be added, the remaining sections must be renumbered to Section 3. Data etc.... Done.

5. Abstract and Conclusion have to be rewritten so to emphasize on the contribution of the study ie 'to examine the role of institutional positions in regime changes.’

We have added the contribution of the paper in the second sentence of the abstract and also revised the conclusion to reflect this suggestion.